# Applied Animal Ethics in Industrial Food Animal Production: Exploring the Role of the Veterinarian

**DOI:** 10.3390/ani12060678

**Published:** 2022-03-08

**Authors:** Elein Hernandez, Pol Llonch, Patricia V. Turner

**Affiliations:** 1Department of Clinical Studies and Surgery, Facultad de Estudios Superiores Cuautitlán, Universidad Nacional Autónoma de México, Km 2.5 Carretera Cuautitlán-Teoloyuca, Cuautitlán Izcallli 54714, Mexico; elein_ht@comunidad.unam.mx; 2Department of Animal and Food Sciences, Universitat Autònoma de Barcelona, 08193 Bellaterra, Spain; pol.llonch@uab.cat; 3Department of Pathobiology, University of Guelph, Guelph, ON N1G 2W1, Canada; 4Global Animal Welfare & Training, Charles River, Wilmington, MA 01887, USA

**Keywords:** animal welfare, animal ethics, food animal, veterinary medicine, sustainability

## Abstract

**Simple Summary:**

Current systems for raising food animals are largely geared to produce large quantities of meat, milk, and eggs, at a low cost to the consumer. There are many ethical challenges associated with these methods, which can result in poor animal welfare and animal suffering. The veterinarian is often undecided as to whom they owe their responsibilities—to the farmer, who pays for their services, or to the animals, who require their advocacy to improve conditions. Historically, veterinarians have focused on enhancing animal health, and have left the ethical debate to others. With increasing consumer attention to animal welfare issues and a global drive to ensure long-term solutions for the planet’s health, it is imperative that veterinarians become more engaged in these ethical discussions. Several examples are provided for considering approaches to some food animal welfare problems.

**Abstract:**

Industrial food animal production practices are efficient for producing large quantities of milk, meat, and eggs for a growing global population, but often result in the need to alter animals to fit a more restricted environment, as well as creating new animal welfare and health problems related to animal confinement in high densities. These practices and methods have become normalized, to the extent that veterinarians and others embedded in these industries rarely question the ethical challenges associated with raising animals in this fashion. Moral ‘lock-in’ is common with those working in food animal industries, as is the feeling that it is impossible to effect meaningful change. Animal welfare issues associated with the industrialization of food animal production are ‘wicked problems’ that require a multi- and transdisciplinary approach. We argue that veterinarians, as expert animal health and welfare advocates, should be critical stakeholders and leaders in discussions with producers and the food animal sector, to look for innovative solutions and technology that will address current and future global sustainability and food security needs. Solutions will necessarily be different in different countries and regions, but ethical issues associated with industrial food animal production practices are universal.

## 1. Introduction

Animal husbandry and management practices have evolved throughout humankind’s recorded history [1]. These agricultural practice transitions have been critical to supporting the growth and expansion of human populations, and have provided food security for growing populations of people in developing urban areas. Most societies that consume meat or other products from animals take a utilitarian ethical approach, accepting that maintaining and killing animals for human consumption benefits society, and harm to animals can be minimized if their needs are met throughout their life, and at the point of death.

The rate at which animal husbandry changes has occurred has accelerated over time, with significant transitions seen in food animal management practices in the late 19th and 20th centuries [1]. The intensification of food animal production began in the poultry sector in the 1930s, initially in the USA, and later in Western Europe, and then expanded to include other species, including pigs, and dairy and beef cows, and other regions of the world [2]. With increasing urban growth and the migration of workers to cities, methods were needed to efficiently and cheaply produce food products from animals, and transport them to consumers before spoilage occurred [3]. Coinciding with the housing of food animals in increased densities, with more confined footprints, was an increase in infectious disease conditions, necessitating the widescale use of antimicrobial agents to preserve the health of animals [4]. Housing large numbers of animals also created a problem of animal waste and the potential for the contamination of groundwater from manure run-off. This complex situation has created a mismatch between societal expectations for animal stewardship (and perhaps idealistic conceptions of how food animals are raised), the reality of on-farm animal welfare, and the efficiencies of scale needed for industrial food animal production.

Veterinarians and animal scientists have been important stakeholders in the growth of the intensification of animal food production throughout the 20th century. Both groups use their knowledge, training, and resources to preserve the health of animals raised for food consumption, while increasing production efficiency. The ethics of housing animals in this manner were not explored deeply, given an animal health-centric view of animal welfare that was prevalent throughout much of the 20th century [5]. In high-income countries, concerns about the conditions and well-being of intensively reared food animals have led to some changes in industry practices; however, more consistent and sustainable approaches are needed as industrial food animal production gears up in low-and middle-income countries to feed growing human populations [6]. The need to better balance human food security and animal production efficiency with animal welfare considerations is not a new concern, being emphasized in 1965 in the Brambell Report [7]. However, the concept has gained recent momentum, as more mature models of animal welfare have been broadly accepted and adopted, such as the five domains model [8,9].

This paper will review the traditional veterinary ethical approach to industrial food animal production, including the normalization of various practices during veterinary training, and contrast this with changes in Western societal ethics and the changing acceptance of food animal management practices. We will also review pertinent animal welfare legislation in the EU and North America for industrial food animal production as it is understood and applied by veterinarians, focusing on pigs, dairy cows, and broiler and layer chickens. This is important because veterinarians are bound to practice obeying local legislation and regulations, but they must think beyond the current state of the industry to support change. Finally, we will explore, through the use of examples, how veterinarians can provide leadership in improving food animal welfare by supporting changes in animal housing and management practices.

## 2. Veterinary Ethics and Intensive Food Animal Production

Industrial food animal production (IFAP) refers to the modern agricultural practices in which fewer farm operations hold increasingly larger populations of animals, often in very dense and confined conditions. On land, these systems are most commonly employed for pigs, and egg and broiler chickens, as well as dairy and beef cattle, but intensive production systems are also commonly used for aquatic food animals. While it is not the intention of producers to harm animals by raising them in these systems, the restriction and confinement inherent in these management systems inhibit natural behaviors, leaving basic drives unmet [10]. Boredom, frustration, exercise restriction, and barren environments may be experienced by animals living in these systems for much of their lives, contributing to poor welfare [11]. In addition, confined space may lead to aggression and other problems that necessitate the surgical alteration of animals, for example, the tail docking of piglets and the castration of most male hoofstock, often without the provision of analgesia or anesthesia [12,13,14,15]. Other public health issues may be associated with IFAP, such as increased disease when animals are kept in crowded conditions, with a resultant need for routine antimicrobial use, and the potential for antimicrobial resistance to develop. In addition, there is a cost to the environment from waste run-off and greenhouse gas emissions, as well as risks to food security [11].

Moral and technological lock-in; that is, the concept that IFAP systems follow paths that are costly and difficult to change, either technically or ethically (because they seem to follow the path of the least bad harms), make it difficult, at least on face value, for industries to innovate [16]. Non-governmental organizations (NGOs) and animal protectionist groups have been significant drivers for changes in IFAP in high-income countries, highlighting systemic problems through public campaigns and undercover videos [17]. These very public exposures may drive change in the private sector, and are often later taken up by the public sector, resulting in legislative changes or restrictions, which require implementation by all [18]. While producer and food consumer groups have often deplored the use of surveillance techniques, significant inertia in the private sector, the difficulty of enacting legislation and regulations, and implied challenges and costs to making changes, have often made both government and industry slow to tackle these problems on their own. That some form of reform is needed for IFAP is not generally questioned by ethicists, but how this may be done is still up for debate [19].

Veterinarians are key stakeholders in IFAP system management, traditionally providing a physical health-based approach to these animals. Veterinarians are well trained to identify physical and physiologic changes in animals, but may not always have the advanced skills or training to cope with ethical dilemmas and to identify and treat animal behavior or welfare issues in practice [20,21,22]. Increasingly, animal welfare content is being strengthened in veterinary curricula [23,24,25,26], but little is known about the application of this knowledge in the field by veterinarians [27]. De Graaf conducted a series of interviews with Dutch food animal practitioners regarding their views on IFAP, and was able to group veterinarians into four categories based on how they conceptualized human clients and animal patients [28]. The groups were: supporters of a responsible farmer; animal advocates; the situational, pragmatic, and intuitive vet; and the professional vet [28]. This study was descriptive only, but it emphasizes that within the veterinary profession, there are diverse moral assumptions and ethical views about IFAP.

While many individuals, including veterinarians and the vast majority of the public, have taken a utilitarian view regarding challenges associated with IFAP in which, on balance, human considerations usually outrank animal ones, this ethical approach does not sufficiently consider the animal welfare or environmental concerns associated with IFAP [29]. In practice, animal suffering has largely been accepted in intensive farming, with few restrictions, other than the most egregious violations (for examples of practices, see: [30,31]). However, the fact that moral distress and ethical conflict are commonly experienced by veterinarians, suggests that this ethical approach may be unsatisfactory for at least some contemporary veterinary professionals, and that other ethical approaches are needed [32,33]. Gjerris et al. suggest that a virtue ethics approach could help to transform the care of food animals, by increasing a sense of attentiveness and responsibility not only toward animals, but also toward the environment that they live in [29]. Rossi and Garner use a common morality ethical approach that is based on nonmaleficence to argue that changes in IFAP systems are needed ethically to protect animals and the environment [34]. A One Welfare framework similarly emphasizes the interconnectedness of humans, animals, and the environment, but focuses on approaches that promote the least bad harms to all three [35]. This framework may represent a more familiar basis for veterinarians to consider their role and responsibilities as they relate to IFAP. This is very similar to a veterinary bioethical approach, which will be discussed further in Section 5.

## 3. Animal Welfare Legislation and Industrial Food Animal Production—Changing Societal Expectations

Worldwide, animal welfare guidelines and laws, when available, have different origins, address different societal interests, and have applied different animal welfare and ethical frameworks. This means that advances in animal welfare practices around the world that rely solely on local or national legislation as the driving force will always be asynchronous (see Table 1 for an example of legislative differences in pig welfare across the top 3 global producers of pigs for consumption).

The development of new welfare legislation takes place in many interconnected and incremental steps. It incorporates information from animal welfare science, social culture, local or national economics, stakeholder lobby group input, and more—as well as requiring the political will to tackle and prioritize. Legislation can take decades to develop, enact and then enforce, and there is also the need to consider the fine balance between making some versus too much progress and expected compliance. Legislation that is too far-reaching is less likely to be successful (i.e., widely respected and upheld) than more moderate legislation, which may also result in the concurrent updating and resetting of social norms [53]. It is important to consider the role of animal welfare legislation as it applies to the ethics of IFAP practices and the practicing veterinarian, because globally, veterinarians are usually considered to be the primary agents responsible for overseeing, enforcing, and implementing animal welfare regulations and guidelines [54,55]. For example, the OIE identifies veterinarians as leading advocates for animal welfare, due to their responsibilities to society, and because of their role in overseeing animal care and health [55]. An understanding of and leadership of animal welfare and related legislation are among the expected specific competencies that veterinarians must have as part of their Day One skills [55]. Furthermore, each national veterinary statutory body and veterinary authority are considered by the OIE to be the competent authority for ensuring the implementation of national and international animal health and welfare measures [56]. Despite this societal expectation, practicing veterinarians are not always closely connected to cutting-edge animal welfare science findings; they are frequently deeply embedded and invested in the food animal production industry as a means of livelihood and self-identity, and they may not feel any conflict with their role or activities with animals, as long as they are meeting accepted practice standards and regulations.

Thus, veterinarians cannot be the only stakeholders with an ethical responsibility to ensure good animal welfare. This responsibility is shared with politicians and government advisors, scientists, and the lay public, who decide what is socially acceptable, in terms of animal care and use. Fisher suggests that by placing the responsibilities for overseeing animal welfare solely on veterinarians and scientists, the relevance of animal welfare as a social construct is ignored [57]. The initial advances in animal welfare protection and legislation were driven by societal ethical concerns regarding animal suffering caused by humans. This eventually led to the creation of non-governmental organizations (NGOs) in the 1800s with animal advocacy and activism interests. NGOs, such as the different branches of the Society for the Prevention of Cruelty to Animals, may have an animal welfare policing role in some jurisdictions through cooperative arrangements with relevant regulatory authorities [58]. NGOs also commonly act as political lobbyists and effect change by raising visibility and public awareness of animal welfare problems in a given country or region [18]. This reinforces that changes in societal ethics can, and do drive changes in global food animal care and management practices and by extension, veterinary practice, even if asynchronous in different regions. This will be further discussed in the following sections, using the EU, North America, and Latin America as examples.

### 3.1. European Legislative Context

In Europe, there has been public debate about food animal welfare issues since the early 1970s. According to the ‘Eurobarometer’ [59], a majority (94%) of EU citizens feel that it is important to protect the welfare of farm animals, and 82% believe that there is a need for the further protection of farm animals. The focus on farm animals is also reflected by increased legislative activity within the EU in the recent years [60], and growth in commercial initiatives led by farmer cooperatives, food animal retailers, and processors [61].

An important event for rethinking food animal welfare in the UK and EU was the publication of Ruth Harrison’s book “Animal Machines” in 1964 [62]. In this book, Harrison describes some realities of industrial food animal production. As a reaction to the book, the UK Parliament requested that the Brambell Committee examine and critique industrial food animal production methods and better define animal welfare. The Brambell Report [7] provided a definition of animal welfare and the first draft of the Five Freedoms, which were later modified by the Farm Animal Welfare Council. The Five Freedoms indicate that farm animals should have freedom from hunger and thirst, freedom from discomfort, freedom from pain, injury and disease, freedom to express normal behavior, and freedom from fear and distress [63]. More recently, the Five Freedoms have evolved into the concept of the Five Domains, described by Mellor et al. [8,9], which incorporate positive mental states as a critical aspect to improve animal welfare. Thus, the Five Domains include the consideration of (1) nutrition, (2) environment, (3) health, (4) behavior, and (5) mental state. The Five Domains model has been adopted and incorporated into various national legislation, and in EU legislation [56], either implicitly or explicitly. Increasingly, there is also a tendency to focus on positive welfare [64,65], such that the next step in the development of legislation of food animals kept by humans is likely to be toward ensuring that these animals have a life that is worth living [66].

### 3.2. EU Legislation Pertaining to Food Animal Welfare

Legislation prohibiting cruelty against animals was first addressed in the UK Parliament in 1822 [56]. The criminalization of cruelty against animals is still today the most common form of animal welfare legislation around the world. Following the UK parliament, a number of countries developed legislation for the prevention of cruelty and unnecessary suffering in the first half of the twentieth century.

Since the 1970s, the EU has incentivized the harmonization of animal protection measures. This process was initiated based on the belief that respect for animals is a common heritage of all European citizens, and that harmonization between countries is necessary [67]. It is also true that harmonization was developed as a means to avoid disparities between national laws to protect animals, which could compromise fair trade within the European common market. Harmonization was initiated through the Amsterdam treaty of 1997, which recognizes that animals are sentient beings and should receive protection against suffering [68]. This was later adopted by the European Council. Five European Conventions lay down the ethical principles driving the use of animals by people [67]. Among them, three concern food animals and two concern research and companion animals.

Given the background of the Amsterdam treaty and the Five Conventions [67,68], the EU Commission has taken the initiative to create legislative texts. Within the European Commission, the General Directorate for the Health and Food Safety (DG-SANTE) has responsibility for protecting animals. When a decision is made about setting up a new piece of legislation to protect animals, DG-SANTE consults a scientific committee (formerly the Scientific Veterinary Committee, which is now a part of the European Food Safety Authority; EFSA). EFSA reviews the scientific evidence for any aspect of a procedure that potentially affects animal welfare and provides recommendations. Following this, DG-SANTE may decide to draft a new legislation, which is submitted to the Council of Ministers of the EU, and it becomes a Council Directive only after receiving their approval. To date, several European directives have been produced that concern all food animal species during rearing, transport or slaughter, as well as some that are species-specific, such as for laying hens, broiler chickens, veal calves, and pigs (see, for example [61,69]).

EU legislation must be translated into national regulations before it can be applied to farms in each country [70]. However, some member states may have their own national legislation that exceeds the European minimum standards. This legislation must, at minimum, conform to European regulations, but may also define more stringent measures. For example, Sweden and Norway have stricter requirements for pigs and prohibit tail docking and teeth clipping, and limit the mandate weaning of piglets to at least 4 weeks [67].

### 3.3. EU Food Animal Welfare Labeling and Assurance Schemes

Today, EU farmers producing animals or animal products for food must comply with national animal welfare legislation, and in addition, they can choose to certify their farm or farm products according to various animal welfare assurance schemes [67]. Veterinarians may be part of this decision-making, but more often, it is the producer making a decision to adhere to a food animal welfare assurance scheme. In 2010, there were at least 67 animal welfare schemes within the EU, with approximately 440 within the food safety area [71]. Due to the rapid development of third-party schemes for animal welfare, the European Commission developed best practice guidelines for animal welfare assessment [72]. Similarly, in the Action Plan on the animal welfare of the European Commission (2006–2010), there was a vision for a market-based approach [73], in which the responsibilities for ensuring good animal welfare shifted from state administration and national ministries to the market place and consumers [74]. In the triad between regulators, consumers, and producers, the veterinarians’ role is seen as facilitating an understanding between the three parties and contributing to higher transparency and fairness in animal production.

The development of a growing market for animal welfare-friendly foodstuffs across Europe is likely to be a key mechanism for further developments in animal welfare standards. The tensions between Europe’s global lead on animal welfare, the realities of World Trade Organisation (WTO) agreements, and the wider global market, have led to alternative market-led initiatives to meet European consumers’ concerns about the treatment of food animals. However, participation in such schemes also reflects differences among farmers in their attitudes and beliefs concerning animal welfare, and is an important motivation for engaging in more animal-friendly production methods. Again, veterinarians can facilitate the harmonization of animal welfare benchmarking, bringing the technical knowledge and objectivity that is necessary to assess and inform about animal welfare.

Finally, in Europe, some schemes operate by including the explicit marketing of better animal welfare. These schemes have been initiated by governments, NGOs, and industry, and by initiatives from food manufacturers, producers, retailers or a consortium. Such schemes may add to and ‘upgrade’ legislation to attract a specific group of consumers [75]. This is the case for the pig sector in the UK, in which farmers have to sign on to a farm assurance scheme (e.g., ‘FreedomFood’ founded by the Royal Society for the Prevention of Cruelty to Animals) that entails stricter regulations for animal welfare, such as restrictions in tail docking [76].

### 3.4. North American Legislative Context

The United States and Canada have dual-federalism systems with a division of powers that permit independent legislation within each state or province that may exceed federal regulations [58]. Although animals are considered property or chattel by criminal courts in both countries, each state or province has a felony animal cruelty law, which defines cruelty, offenses and prohibitions, and penalties [58,77]. However, food animals are not specifically considered in all state or provincial animal protection legislation. Whiting suggests that this is due to a long-term social acceptance of farming practices, which has made it challenging to increase veterinary or other surveillance of farm animal practices. This has led to people adopting an ‘all or nothing’ approach to accepting animal agricultural practices in North America [78]. Most jurisdictions specify the duties and penalties of the people who are responsible for food animals. California’s Penal Code includes “every owner, driver, or keeper of any animal without proper care and attention as guilty of a misdemeanor” and describes the responsibilities of “peace officers, humane society officers, and animal control officers” [79]. In Canada, the British Columbia Society for the Prevention of Cruelty to Animals Act describes that the duties of the person responsible are “to care for the animal, including protecting the animal from circumstances that are likely to cause distress” [77,80]. These examples highlight some of the subtleties in wording and ambiguous scope used in various provincial and state legislation that make it difficult for veterinarians to take a consistent approach for enhancing the care of animals, especially if moving between practices in different provinces or states.

In both the USA and Canada, veterinary practitioners are responsible for providing animal welfare education to farmers and producers, together with marketing boards and agencies, and for certifying services related to animal or food sanitary status [58]. In the USA, veterinarians may participate in animal health and welfare policy programs through the United States Animal Health Association (USAHA) [81,82]. In addition, within the American Veterinary Medical Association (AVMA), the Animal Welfare Division has the mission of assisting and overseeing human–animal interactions [83]. Most animal protection law is developed at the provincial or territory level, and is not of federal origin in Canada. This policy framework development is considered incidental and likely a secondary result of the need to address particular issues at certain times [77]. Food animal welfare issues have been addressed nationally by national codes of practice developed and supported by the National Farm Animal Care Council in conjunction with various stakeholders, including retailer, marketing, veterinary, and producer groups [84]. The codes are considered a minimum acceptable standard of care for the various species covered, and have been written into law in some provinces. Similar to the AVMA, the Canadian Veterinary Medical Association (CVMA) has an Animal Welfare Committee that develops standards, guidelines, and position statements concerning the care and well-being of food animals and other animals [85]. This guidance can be used to support ethical decision-making by veterinarians.

### 3.5. Food Animal Welfare Assurance Schemes in North America

Similar voluntary third-party food animal welfare inspection and assurance schemes exist in North America as for the EU. In contrast to European animal welfare assessment, participation and compliance with recommendations may be mandatory for some producers based on buyer or retailer requirements and has developed in a heterogeneous way by including species and farming system-specific assessments and having different legislative scopes within North America. Many buyers of food animals or animal products also conduct internal welfare assurance audits at farms to manage risk associated with animal welfare issues and adverse publicity [86,87]. In the US, the United Egg Producers was one of the first food producer groups to develop animal welfare standards and third-party auditing programs likely secondary to a campaign against induced molting. Similarly, other retailers (e.g., McDonald’s, American Meat Institute, Food Marketing Institute, National Council of Chain Restaurants) have developed voluntary standards and auditing guidelines [88]. Assurance schemes define acceptable animal welfare standards and encourage producers and veterinarians to exceed the minimum requirements of care expected for food animals under legislation. They may include general guidelines for improved food animal housing and husbandry, refined animal handling and restraint, and acceptable somatic cell count and mortality levels [88]. They are important for raising animal welfare standards between legislative changes and can be used by veterinarians to encourage improvements in animal care.

### 3.6. Latin American Context

Latin America (LA) comprises several countries with different agricultural conditions and interests. The LA food animal production sector has had a higher annual growth rate (3.7%) than the average global growth rate (2.1%) [89]. Therefore, assurance of good food animal welfare and production practices are a relatively new concept for this region, but of increasing interest due to international trade and animal movement policies [90]. However, several countries have yet to implement specific food animal welfare policies. Glass et al. surveyed 22 countries to assess the awareness level of OIE animal welfare standards in LA [91]. They identified that 28% (7/22) of countries have no animal welfare policies covering food animal transportation by land. In addition, 18 respondents with specific or partial national standards did not cover all 12 elements of the OIE Terrestrial Code [91]. This makes it difficult for progressive veterinarians working in this region to push for changes in food animal care and management.

Most LA countries have more specific national or local policies about animal cruelty or husbandry practices (i.e., transport and humane killing of food animals), which are predominantly enforced by national veterinary authorities [90]. This disproportionate development of animal welfare frameworks amongst LA countries is likely secondary to economic interests rather than for ethical reasons, compared to European animal welfare legislation [92]. The lack of formal animal welfare regulation also reflects the lack of formal veterinary training in animal welfare and applied ethics [93,94]. Animal welfare training mainly focuses on veterinarians and veterinary students with a smaller portion of education targeted towards paraprofessionals working in the agricultural sector [91], because veterinarians are considered the competent authorities for enforcement of animal welfare policies in LA [90]. Unlike the situation in the USA and Canada, most LA countries have not developed a division focused on animal welfare within their professional veterinary associations. There is an international, collaborative support system for the veterinary profession within LA regarding animal welfare that provides training and guidance on some animal welfare policies. The OIE has appointed a Focal Point for Animal Welfare to promote the development and dissemination of knowledge in each American member country [95]. In addition, there is a joint effort from the veterinary schools in Chile, Uruguay, and Mexico, as well as other countries, to support the OIE Collaborating Centre for Animal Welfare Research and Livestock Production Systems. The center’s mission is to promote and provide veterinary and scientific expertise on legislation, research, and information on good animal handling practices in livestock systems in the Americas, particularly in LA [90,91,96].

This review of legislative frameworks within the EU and North and Latin America suggests different motivations for developing animal welfare policies and approaches. Policies and practices in North and Latin America are less driven by societal pressures and a utilitarian human-focused ethical approach predominates. This underpins veterinary attitudes towards food animal welfare and emphasizes that veterinarians and others must employ different strategies to advance farm animal welfare even within high income countries around the world.

## 4. Applied Ethical Challenges in Modern Industrial Animal Farming

In 2012, Dawkins emphasized that animal welfare had been pushed off relevant farming and agriculture political agendas despite, or perhaps because of, the increasing recognition of the need for efficient food production for an estimated 9 billion global population by 2050 [97]. Subsequently, the United Nations Committee on World Food Security included animal welfare among the specific challenges of intensive livestock systems [98,99]. This also was emphasized in 2021 as part of the One Health, One Welfare initiative. The 2016 report acknowledges the challenges for intensive industrial systems balancing between an increase in production and growing demands for efficiency and welfare improvement [98]. The concept of intensification of food animal practices has a strong negative connotation for those interested in animal welfare because it is associated with reduced freedom to express natural behavior and, likely, the opportunity for a good life [100]. However, not all food animal intensification practices necessarily reduce animal welfare, and some provide opportunities for increasing the efficiency of husbandry practices and provision of animal care. As Buller et. al. note, animal welfare science cannot provide unconditional support of less intensive and organic systems and reject intensification based only on animal welfare and advocacy without considering the collateral food safety and security issues [101]. Animal welfare science instead has to provide the methods to objectively assess animal welfare, so that decisions can be taken based on objective criteria. The role of veterinarians should be to transfer this knowledge to farmers so they can monitor the status of their animals and act accordingly, regardless of the production system. The ethical dilemma of intensification will continue to be a challenge for veterinary professionals as they are constantly required to consider their client’s interests, public opinion, and their own ethical framework when making decisions concerning animal welfare.

Animal farming must shift towards more sustainable intensification systems, but there is still great controversy defining what is meant by sustainability. By definition, sustainable systems consider the use of world food resources and the long-term effect of different practices on environment, animal welfare, and human health, with an eye to balancing outcomes [102]. Sustainability requires a multidisciplinary approach involving environmental, ethical, social, veterinary, and economic stakeholders to solve issues [100]. There is unlikely to be a single solution across low-, middle- and high-income countries because of an imbalance in resources and needs [91]. Animal science has provided solutions using multidisciplinary approaches for specific animal welfare issues (e.g., the use of appropriate colostrum management programs as a replacement solution for individually housed dairy calves to reduce disease transmission) [103]. However, there are no immediate solutions for the issues associated with intensification of food animal practices, and it has been suggested that more radical actions must be taken as “low-hanging fruit” solutions are not the solution for long-term change [101].

Further veterinary understanding of the background behind routine farming practices is needed to adapt today’s common practices into tomorrow’s modern moral thinking. There is a normalization of various practices during veterinary training and in practice that challenge sthe four ethical principles of autonomy, beneficence, non- maleficence, and justice in IFAP [104]. For example, the humane killing of male layer chicks and bull dairy calves are widely accepted in the poultry and dairy industries, respectively, because these animals do not have an economic value [31,105]. However, the beneficence and non-maleficence principles of veterinary professional ethics (i.e., to promote benefit for the patient; to avoid harm, respectively) are challenged by these practices. Similarly, food animal veterinarians encounter scenarios in which it is difficult to apply justice and autonomy (i.e., to provide benefits, risks, and costs fairly; to respect the patient’s own decision regarding their care) to their patients (e.g., treatments that are refused by owners due to economic constraints) and this has become acceptable out of economic necessity in veterinary medicine [106]. These values and issues differ between cultures. Ultimately, society takes a utilitarian view based on the food safety and a desire for low food prices secondary. More profound trans- and multidisciplinary changes are needed if veterinary medicine is to adapt to support sustainable solutions for IFAP.

Aligned with other ethical and welfare concerns associated with confinement of agricultural animals is the use of genetically modified (GM) animals. GM animals have been developed in research settings to improve disease resistance, meat and product quality, animal productivity, and environmental sustainability [107,108]. Different techniques involve adding, removing, or changing specific DNA sequences with precise gene-editing tools. Precise gene-editing approaches have been suggested to improve animal welfare as an alternative to conducting painful management practices like dairy calf dehorning and pig castration [109]. DNA-selection tools could also be more economically feasible compared to traditional crossbreeding techniques [109]. Other GM methodologies have been tested to minimize the environmental impact of large animals such as pigs. The best-known example is the Enviropig project that created pigs expressing salivary phytase, which were supposed to enhance phosphate use in animal-fed cereals, thereby reducing environmental pollution by producing 60% less fecal phosphate [107].

However, there are several animal welfare and ethical concerns for veterinarians associated with the production of genetically engineered animals. Until relatively recently, researchers’ main limitations were the resources or technology needed for research, but now researchers and veterinarians also must consider the ethical and societal acceptability of their studies [108,110]. Kasier has argued that the ethical concerns about animals in biotechnology are either intrinsic (based on the means of production) or extrinsic (based on its consequences), and the effects on animal welfare require a thorough assessment on a case-by-case scenario [111]. For example, the Enviropig project objective was to address environmental pollution issues. An intrinsic approach could justify that this was morally problematic because it is ‘unnatural’ to genetically modify animals. In animal agriculture, there is a strong gut feeling that processes need to be ‘natural’ despite over 3000 years of domestication and selection for genetic traits and characteristics. Veterinarians who apply ethical thinking in their daily practice might question the significance of ‘unnatural’ selection and their responsibility for enhancing animal welfare [112]. Similar to the previous dilemma, more proactive risk assessments need to be carried out for each scenario, considering the moral weight of the potential uncertainties [111]. In 2012, the Enviropig pig work was ultimately cancelled, but it raised several extrinsic ethical considerations including perpetuation of intensive pig farming and monopolized corporate agriculture [113]. Veterinarians need to speak up as animal welfare advocates using scientific and ethic-based principles rather than react defensively to constantly changing public expectations and moral pluralism. This proactive animal welfare risk assessment should also be used during the development of technologies that may raise ethical concerns viewed from some stakeholders’ perspectives (e.g., precision livestock farming, with the use of sensors to measure and analyze large resource and animal-based data that aids producers but also promotes increased intensive farming and less holistic observation of animals [114]).

Another challenge that veterinarians may face is justifying their stance within multi- and transdisciplinary initiatives. The One Health [115] and One Welfare [116] frameworks encompass factors affecting humans, animals, and the environment [116,117]. These frameworks endorse the concept that the veterinarian’s role and responsibilities are not only associated with animal health but also with public health, human and animal welfare, and environmental health. Although the concepts are not new, they are intended to strengthen “the collaborative efforts of multiple disciplines working locally, nationally, and globally, to attain optimal health for people, animals, and our environment” [118]. These initiatives highlight that veterinarians must not limit their attention to an already potentially complicated client-patient-veterinarian relationship and should also consider their responsibility to broader imperative, namely public health, animal health and welfare, and ecosystem health. These topics are embedded in the oath of veterinary professionals around the world. There is an opportunity for veterinarians to bring the animal health and welfare perspective into the debate and clarify controversial animal welfare issues (e.g., zoonotic diseases, food security, antimicrobial reduction, etc.). Bioethicists have discussed some pitfalls of and concerns about the One Health concept by describing the ethical and normative implications of acknowledging these elements and their relationship into our moral belief by implying that even the “non-living environment” has a moral status [119,120]. However, it is also time for veterinarians and animal welfare scientists to engage in a multidisciplinary bioethical approach when considering IFAP challenges.

## 5. The Veterinarian’s Role in Ethical Decision-Making—Examples

In this section, several IFAP scenarios will be discussed with consideration as to how veterinarians could approach these issues ethically to challenge the status quo. As noted in Figure 1, veterinary ethical decision-making follows a common pathway that incorporates identifying the problem, defining stakeholder perspectives, reflecting on personal values, applying existing legislation, analyzing the consequences of choosing alternative actions and taking action, regardless of the ethical framework used. An applied veterinary bioethical approach has been used to describe approaches and potential future steps toward tackling the issue. Veterinary bioethics has a broader scope than veterinary medical ethics, being applied to evaluate the treatment and care of animals, in setting optimal welfare standards, and in determining what is best for animals from the perspectives of medicine (i.e., disease prevention and treatment) and ethology. Veterinary bioethics, in its broadest consideration, applies to animal health and well-being, environmental health and quality and the animal impact on this, and human interests and responsibilities [121]. This type of approach also considers the consequences of various initiatives, going beyond a simple harm: benefit analysis (consequentialist approach). Whereas veterinary medical ethics argues for the best scientific knowledge and medical and surgical expertise to be provided to animals at the owner’s request and payment for professional services (i.e., a more ‘service-based’ approach), veterinary bioethics has a mandate to evaluate the client’s request in terms of the patient’s best interests—potentially a more holistic approach [121].

Scenario 1: Use of gestation stalls on pig farms

Background: Intensive food animal production protects livestock from environmental extremes and predators, potentially facilitating closer and more frequent inspection by stockpersons, leading to better nutritional and health management. Despite this, there are costs to the animal, including impaired social behavior, limited choice and control within the housing environment, poor environmental stimulation and restrictions in performing natural behaviors. Although intensive farming has revolutionized the availability and affordability of animal protein, there are many societal concerns concerning its ethical implications including associated animal welfare threats.

Meat from pigs is the greatest animal protein source consumed around the globe compared to other terrestrial animals [122,123]. Specifically, the FAO reported that of global meat consumption, 37% is pork (110 million metric tonnes, mmt), well ahead of beef (67 mmt) and chicken (104 mmt). To produce this quantity of pork, pig farming has undergone profound industrialization with a significant intensification of animal breeding and housing. This includes an optimization of space and confinement of animals in very high density. In pig production, one of the confinement practices designed to improve efficiency is the accommodation of pregnant sows in gestation stalls.

Gestation stalls allow individual animal feeding and management and minimize overt inter-animal aggression. However, the severe restriction of movement and the inability to perform normal feeding and social behaviors can lead to welfare problems such as development of stereotypies, lameness, and decubital ulcers, among others [124].

Restriction of social contact and movement during gestation conflicts with innate motivations of pigs, resulting in frustration that compromises their welfare. Also, such restrictive confinement prevents postural changes and wallowing, both significant means for pig thermoregulation. In addition [125], close confinement results in other general welfare concerns, such as a stress reaction to reduced cardiovascular fitness [126], impaired bone strength [127] and increased morbidity [128]. Sows are also commonly feed restricted in these settings, facilitating the development of oral stereotypies [129].

Group housing of gestating sows may address many of these welfare problems. In these settings, sows are allowed to express their natural behavior during estrus, better explore their environment (providing for better choice and control) and change lying posture and wallowing to thermoregulate. The optimal time for grouping may be immediately after weaning as social acceptance increases during estrus [130].

Following the recommendations of the scientific community and NGO lobbyists, some western countries have limited the use of stalls for gestating sows, with the aim of improving animal welfare. For example, since 2013, the EU has banned the use of individual stalls for gestating sows. Pregnant sows must be group-housed from day 29 of pregnancy to one week prior to parturition. Similarly, some large retail firms that sell pork in Australia, Europe, Canada and the USA have issued statements indicating that they will only source pork from farms that do not use gestation crates [131,132].

Veterinary and producer responsibilities: Group housing of sows has some disadvantages. Individual feeding and supervision is more difficult, although the main welfare problem are injuries and stress caused by aggression, particularly after mixing animals that are vying for limited resources such as feed. As noted by the EU Scientific Veterinary Committee, it is important that group-housing systems are adequately designed and managed so that welfare remains acceptable [51]. There is a complexity of design and management practices of commercial group-housing systems, although many of these features can affect sow welfare. Housing considerations, such as space allowance, group size, maintenance of static and dynamic groups and use of a mixing pen, and sow nutrition, including diet, ration, and feeding system, all may affect sow welfare. Also, sow welfare may be influenced by animal characteristics, such as genetics, experience, stage of reproduction, and parity, as well as by quality of stockmanship. Given this wide disparity of variables, the farmer and veterinary practitioner play key roles. The farmer has the responsibility to provide adequate care to animals. Individual care will likely benefit each animal as it will refine the provisions to the individual needs. Monitoring the individual animal requires commitment from farm staff. The veterinary practitioner is responsible to provide husbandry, management, and care recommendations to each farmer, considering the intrinsic factors of each farm, finally adapting the recommendations to the context. For instance, feed provision for farms that have a large sow turnover may facilitate grouping sows in more homogenous groups (in terms of body size and parity), which may help to reduce overall aggression. This may be different for smaller farms, in which sow groups may be more heterogeneous. Therefore, the skills and knowledge of both farmer and the veterinary practitioner must adapt to these circumstances.

However, the decisions of the farmers and veterinarians also should take into account the economic sustainability of the farm. According to Edwards, the extent to which acceptable economic performance can be realized with alternatives to gestation stalls depends on the relationship between performance and the cost requirement, (i.e., the inputs vs. outputs of the system) [133]. An important issue to consider is the initial investment to cover the installation of new pens, etc. In this regard, Schulz and Tonsor review several studies that estimated the direct costs of switching from gestation stalls to group pen housings [132]. According to them, there is a general agreement of increasing costs at the farm level, but the magnitude of this increase is highly debated.

Another critical aspect, which is important for long-term sustainability is the revenue-generating capacity of the system, which relates to the level of animal reproductive performance relative to the variable cost requirement. Some authors have addressed this point and show that well-managed gestation stalls and group housing conditions produce similar outcomes in terms of physiology, behavior, performance, and health [131,134].

Poor economic return would make a producer less competitive and at risk of bankruptcy if they use a group housing system that requires more floor space. One cannot predict the future with certainty and animal production may be affected by many unexpected events (e.g., China’s pork crisis after African swine fever decimated a significant portion of pig farms). Despite this, as the global population increases, consumers from middle- and high-income countries may be willing to pay a bit more for pork raised in a sustainable manner or to consume less meat as part of a healthier lifestyle. When possible, improved welfare standards should be rewarded financially, so those producers can support investments and innovations to turn their business towards more animal welfare-friendly systems.

Ethical decision-making: Animal welfare is a contentious and emotional issue in livestock production and IFAP, and this includes the use of gestation stalls for sows. The welfare of livestock in commercial production systems has received attention from many groups, including consumers, animal protectionist groups, scientists, legislators, veterinarians, and producers.

A more enlightened consequentialist perspective has replaced a largely anthropocentric perspective amongst pig producers over the years, in which the interests of the sows matter and their lives are not only considered as a means to an end. Despite this, most pig farmers and veterinarians working in IFAP systems do not necessarily question the right of humans to produce pigs or keep sows in gestation stalls. Regardless of animal welfare advances in guidelines and legislation for the use or ban of gestation stalls, veterinarians must address ethical dilemmas. An ethical conflict arises between the interests of the animal (partially supported by societal expectations) and the producer’s interests, who pays for treatment. Legally, in most jurisdictions, the animal is the property of the owner. Therefore, the owner may prefer some farming methods that their veterinarian disagrees with. In such a scenario, the dilemma is whether to maintain (or treat) animals according to the owner’s expectations or to proceed with practices that the veterinary practitioner believes are appropriate. According to Rollin, this is the fundamental question of veterinary ethics: “to whom does the veterinarian owe their primary obligation—the owner or animal?” [121,135]. In an anthropocentric model, the animal’s needs are not directly considered, whereas an alternative model (that Rollin calls the ‘pediatrician model’) would primarily focus on the welfare of the animal in any decision over its treatment [135]. Similarly, de Graaf (2005) noted that some veterinarians might accept IFAP practices (e.g., the use of gestation stalls) if their interests are aligned with the producers, in which the sows are considered economic units. This discourse is described as ‘veterinarians that support of the responsible farmer’. Whereas an ‘animal advocate veterinarian; might disagree with current IFAP practices, including gestation stalls [28].

Despite these different veterinary perspectives, veterinarians working in industrial food animal practice may be able to gain traction for change through work with producer and veterinary medical associations, knowledge of small business or agricultural improvement loans programs, and regional or national marketing and standard-setting groups. Consideration of long-term sustainability of practices should drive producers and veterinarians towards common solutions, even if for different reasons.

Alternatively, a One Welfare approach could be used to address the conflict between animal welfare and efficiency/productivity in IFAP systems based on the improvement of human-animal relationships. Labor satisfaction and performance (i.e., related to producer well-being) could improve by acknowledging that animal welfare issues secondary to IFAP practices are being addressed on the farm (e.g., the use of group of gestation stalls regardless of the economic or management issues they imply) [35]. There is a need for more research focused on the human-animal relationship and its relationship with productivity in pig farms.

Scenario 2: Greenhouse gas emissions as a by-product of food animal production.

Background: Food animal production contributes significantly to greenhouse gas (GHG) emissions. The two main GHGs associated with animal production are methane (CH_4_) and nitrous oxide (N_2_O). Methane is produced during enteric fermentation, whereas N_2_O arises from transformations during manure management as well as deposition of animal manure on pastures [136]. In 2010, methane was estimated to be responsible for more than 30% of global livestock emissions (CO_2_-eq/year) followed by nitrous oxide (N_2_O) (~20%) [137,138].

Because of a forecasted increase in the consumption of animal products, livestock producers should consider means to reduce the impact of animal production on the environment. Some industrialized aspects of food animal management, including nutritional and genetic improvements, may support mitigation of GHG emissions, especially in less intensive production systems, where there is still much room for improvement, thereby reducing emissions intensity (Ei) and improving animal welfare at the same time [139].

Emission intensity is a measure of the quantity of GHG emissions generated per unit of output. Different strategies are based on increasing production efficiency, seeking to reduce GHG emissions while maintaining the level of production. An example of an important strategy to reduce Ei is improvement of the health status within a herd, which not only improves the environmental sustainability but also animal welfare and economic return at the same time, acting as win–win-win strategy.

Emission intensity is inversely associated with productivity of a system, measured in terms of output per animal or the herd. This is based on evidence that more efficient systems create less waste, including GHGs, per unit of product [140]. For example, increasing livestock efficiency would require fewer animals as well as shorter lifetimes to produce the same quantity of product. It results in a reduction of the inputs necessary for production and associated waste [141]. The Ei mitigation approach allows reduction of GHG emissions and increased profitability at the same time. Yet, a drive for improved system efficiency has driven livestock intensification (e.g., restricted grazing, breeding for high producing animals, etc.) which, if a certain threshold is exceeded, may go against the welfare of animals (e.g., increasing stocking density).

Animal welfare is a necessary element of sustainable animal production [142], which is increasingly demanded by society [143]. This is also acknowledged by the UN Committee on World Food Security, in its ‘Proposed draft recommendations on sustainable agricultural development for food security and nutrition including the role of livestock’. Recommendation ‘D’ of Article VIII, identifies animal welfare as a distinct component of sustainable agricultural and economic development, of food security, and of human nutrition [144]. A growing number of consumers demand ethical animal production and refuse products that they consider to have been produced under circumstances that are morally unacceptable [142]. In one study, 88% of consumers indicated that they used animal welfare as a choice criterion [145]. Whether consumers act as virtuously as they indicate outside the parameters of a study is unknown. But this does suggest that there is public interest in ensuring animal welfare is improved at the same time that climate change mitigation measures are implemented in animal agriculture.

Veterinary responsibilities: Good standards of animal welfare cannot be achieved under conditions of poor health [146]. Poor livestock health is associated with behavioral and metabolic changes, such as reduced feed intake, a reduction in feed digestibility and increased energy requirements for maintenance [147]. Altogether, these costs of disease may lead to inefficiencies and a reduction of growth capacity that in turn raises GHG Ei [141]. As an example, in dairy cattle, both lameness and mastitis, reduce milk output, increasing GHG emissions per litre of milk produced [148]. Conversely, improvements in health may reduce inefficiencies from poorer productivity of individual animals but also from product condemnation [149,150]. Better health may reduce culling due to injury and disease, and is very likely to extend the average productive life span of the herd.

Improved animal health through the prevention and control of disease and parasites is widely regarded as fundamental to animal welfare [151]. The role of the veterinary practitioner in maintenance of good herd health is paramount. Veterinarians around the world should continue to focus on prevention and treatment of diseases to improve animal health. In this regard, efforts on improving the health status of the herd will result always in a better production efficiency, therefore will facilitate the reduction of environmental impact of animal production.

Animal welfare is not only determined by health, but also non-health aspects such as comfort, absence of fear and the ability to perform natural behaviors. In some circumstances, better animal welfare can benefit productivity and thus reduce GHG Ei [139]. For example, stress associated with negative handling can reduce milk and meat production in dairy [152] and beef cattle [153]. In laying hens, too high a stocking density can lead to a reduction in productivity [154]. Similarly, in pigs, the growth rate of pigs subjected to restricted space allowance can be depressed up to 16% [155].

Ethical decision-making and dilemmas: The above scenario exemplifies strategies to complement two main societal goals such as environmental sustainability and humane production as win-win strategies. In such conditions, no ethical dilemma exists and tackling one problem provides solutions for different problems. However, in other cases, an ethical dilemma exists when the commitment to improve the welfare status of animals is in contradiction to improvements in environmental sustainability. Farmers are moved by society to become more efficient and this potentially puts animal welfare on a collision course with profitability [156]. For example, animal welfare is higher when animals are allowed to perform natural behaviours. In ruminants, grazing is a behaviour considered to be highly natural and for which cattle show a strong motivation. However, it is also true that grazing may, in some cases, increase the ratio of GHG emissions (i.e., N_2_O) compared to confinement systems in which manure is stored and managed differently [157]. An obvious means of improving farming efficiency is to increase the number of animals kept on a farm as there are less indirect environmental costs per animal as stocking density increases [158]. Increased stocking density is associated with negative welfare effects for many species [159]. In dairy cows, genetic selection for increased milk production has led to an increasing incidence of health problems, such as lameness and a decline in longevity and fertility [160].

The food animal practitioner may face a dilemma regarding whether to promote the welfare of animals or reduce environmental pollution. Protecting the well-being of the animals or patients is within the code of conduct of the veterinary practitioner and within their ethical obligations. On the other hand, protection of the environment is driven by general interests of society. In this context, environmental motivations may be less prioritized in the face of economics, supported by producers, and animal welfare, supported by veterinarians. Therefore, measures to diminish the impact of livestock on the environment could receive support by policymakers by creating a legislative framework that facilitates the implementation of these measures. In any case, the decision as to which measures should be prioritized and by how much, is not only responsibility of the veterinary practitioner or the producer but is also incumbent on society.

Scenario 3: Reduction of antimicrobial use in poultry production

Background: Antimicrobials have been used widely and for many decades for therapeutic and disease prevention properties and as antimicrobial growth promoters (AGP) at sub-therapeutic doses to increase efficiency in various animal production systems. Over the past 50 years, the use of antimicrobials has helped the poultry industry to treat and prevent diseases [161]. In addition to other external factors (e.g., rearing conditions, genetics, husbandry practices, etc.), the administration of antimicrobials modulates the intestinal microbiota of poultry and hence their immunity and health [162]. The mechanism of action of antimicrobials depends on the group of antibiotics used and can be administered via drinking water, in-feed medications, or injectable administration. Poultry are highly susceptible to vertical and horizontal transmission of extra-intestinal pathogenic bacterial strains that have been commonly treated with a prophylactic approach using medical antibiotics for humans. Antimicrobials have been a practical solution for controlling enteric diseases and sepsis in poultry.

Over the past several decades, intensive animal production has created concerns associated with overuse of antimicrobials. In poultry, widespread antimicrobial use has also been associated with bioresistance and drug residues in poultry products worldwide [162,163,164]. This represents a public health threat due to the potential of transmission of antimicrobial resistance (AMR) from animals to humans because of poorly regulated antimicrobial usage in the veterinary medicine [165]. This has created a significant public health threat because >50% of the total annual production of antimicrobials is used in veterinary medicine [166] and poultry production is among the fastest-growing industries in the world.

Globally, different legislative approaches have been developed, including banning antimicrobials for growth promotion [167], reduced used [168,169] or introducing a special tax on meat from animals treated with antimicrobials [170,171]. The tripartite (FAO-OIE-WHO) considers AMR among its top three One Health priorities [172]. In Europe, the use of AGP has been banned since 2006 through a combination of voluntary and mandatory legislative strategies [167,173], resulting in a marked decline in the use of antimicrobials in food animals [174]. In the USA, there are federal recommendations for the appropriate or judicious use of antimicrobial drugs in food-producing animals [175]. However, whether national and international policy development can effectively reduce the development of AMR in animals is controversial. In 2005, fluoroquinolone use in U.S. poultry farms was banned to reduce the prevalence of fluoroquinolone-resistant *Campylobacter* spp., but reports exist indicating persistence of AMR bacteria after its use was discontinued in chicken carcasses [176,177]. Conversely, the ban on the use of avoparcin and virginiamycin in Denmark was followed by a decrease in the occurrence of AMR bacteria in broiler chickens [161]. Despite policies to withdraw or restrict antimicrobial use in food animals, there is some evidence of increased therapeutic drug antimicrobial resistance [178,179,180,181].

The use of antimicrobials is also a concern in most low- and middle-income countries because broiler chickens are the farm animal species with the highest antimicrobial use, followed by pigs [182]. China (a middle-income country) is the main producer and user of antibiotics in the world and has reported the misuse of antibiotics in intensively farmed poultry despite regulatory efforts to control use [183]. Similar findings have been reported in other low- and middle-income countries with limited regulatory effect due to weak regulatory oversight and/or a disorganized production and marketing system [184,185].

Veterinary responsibilities: Farmers and veterinarians are generally highly knowledgeable individuals regarding poultry production, but a lack of focused training for producers coupled with ready drug availability has resulted in misuse. Consequently, the public has looked negatively upon antimicrobial usage without considering the health implications to animals if they are left untreated [186].

In the poultry industry, several countries have feared banning of AGP and the associated economic implications, because many studies have focused on the economic consequences of bans for growth promotion and not on the dramatic reduction of antimicrobial use [168,169,187]. With a more holistic approach towards good husbandry, hygiene, and improvement of biosecurity practices there can be significant health, welfare, and economic benefits that exceed those seen that follow the routine use of antimicrobials [167,171].

Veterinarians are stewards in prescribing antimicrobials and must understand that the consequences of prescribing antimicrobials goes well beyond the specific treated food animal. In contrast to companion animal medicine, veterinary practitioners in IFAP systems deal with populations of animals when controlling infections and diseases. Not only are these veterinarians responsible for ensuring the health and welfare of poultry but also for protecting public health and food safety.

Different regulatory and monitoring bodies have highlighted the importance of disease prevention strategies to reinforce prudent use of antimicrobial and reporting of AMR and residues [188,189]. Complete abolition of antimicrobials on farms could threaten the interests of animals, producers, and the veterinary profession, and thwart bioethical principles of good care. Consumers also must acknowledge that antimicrobial use cannot be completely avoided. Veterinary practitioners are ethically obliged to attend to the principle of beneficence and provide veterinary care to ensure good welfare of the animals. This could be achieved by responsible administration of antibiotics, as appropriate. Veterinarians must follow antimicrobial use guidelines by limiting use to that which is strictly necessary and to provide improved oversight and training for their use in food animals, including providing estimated meat and milk withdrawal times [188]. Development of AMR may result in animal welfare issues associated with pain, discomfort, poor growth or weight gain, and productivity changes, such as increased culling [190].

Ethical decision-making and dilemmas: In this scenario, there are multiple ethical issues that the veterinarian may identify and these may represent causes of moral stress. A One Welfare/One Health approach can be used to highlight the importance of these therapeutics in human and veterinary medicine. Traditionally, antimicrobial use has been considered a cheap gateway to continuing unsustainable practices instead of seeking more long-term sustainable solutions to better care for animals and minimize disease [167]. Veterinarians play a unique and occasionally morally conflicting role by having to defend public health and by respecting the client’s interests, especially if they differ from the veterinarian’s opinion. The veterinarian’s actions also strongly depend on the local regulatory framework regarding antimicrobial use. Good communication about risk analyses and animal welfare issues is needed to overcome contrary perspectives. There is no ethical dilemma if these conditions are achieved. However, an ethical dilemma exists when the commitment to improve the welfare status of animals contradicts current policies or stakeholder principles.

Page et al. suggested a ‘5Rs’ approach for antimicrobial stewardship. In addition to the ‘3Rs’ (i.e., refinement, reduction, and replacement), the authors propose ‘responsibility’ and ‘review’ [191]. This approach provides the veterinary practitioner with guidance on decision-making when prescribing antibiotics according to current practices and policies (‘review’) and accepting the potential consequences (‘responsibility’). In addition, the veterinarian should acknowledge that antimicrobial use should generally be reduced, be specific for each treatment, and be replaced when evidence supports the use of alternative measures (i.e., reduction, refinement, and replacement, respectively) [191].

Efforts to harmonize control and use of antimicrobials in food animals have not been successful within and between countries. Few studies have been undertaken to understand the social and economic drivers behind antimicrobial use in the poultry industry [185]. In some countries the use of ‘last resort’ antibiotics in poultry remains underreported and has likely increased the environmental burden of AMR [182,184]. There is a need for more preventive management strategies within the poultry industry and heightened antimicrobial stewardship.

Scenario 4: Ethical decision-making process for euthanasia of lame dairy cows

Background: Intensive farming systems make use of higher stocking densities as well as housing conditions that may limit a cow’s natural expression of behavior. Lameness is a common, multifactorial condition and is a major animal welfare and health issue in dairy cattle. The condition causes pain, negative affective states, changes in productivity (e.g., decreased fertility and milk production, increases risks of other diseases), and increases culling risks [192]. Veterinarians and producers must be able to identify the early stages of the disease to minimize poor welfare and seek treatment. Failure to treat cows can be caused by producers and veterinarians underestimating the problem, economic concerns regarding veterinary treatment, or a lack of understanding of pain management [193]. The estimated financial losses associated with lameness are approximately $378 USD per case with sole ulcer and foot rot being the most significant problems [194]. However, this does not consider the welfare cost to the animal. In addition to pain, lameness affects the animal’s ability to perform social interactions, and alters resting behaviours, and, potentially, digestive functions [195].

Despite shared concerns between stakeholders about the importance of recognizing and treating lameness in the dairy industry, there is a lack of agreement on its definition and normalization of signs of animal suffering [196]. The complex nature of this disease encompasses the issue of an adequate definition of lameness among stakeholders and causes frustration in veterinarians who may be trying to provide treatment [196]. This misalignment in objectives can later result in increased culling rates secondary to poor animal management.

Veterinary responsibilities: Similar to the previous scenarios, veterinarians are accountable to multiple stakeholders (see Figure 1). The veterinarian has a responsibility to the patient but also to the owner. Despite lameness being a common issue within the dairy industry, veterinarians can face challenging situations with producers in which it may be difficult to provide simple messages, and building trust and receptivity to messaging takes time [196]. Veterinary practitioners must act in a way that demonstrates an understanding of their ethical and legal responsibilities. Veterinary practitioners should prioritize lameness-related pain recognition and treatment—this will improve productivity and overall farm economics and reduce mastitis from cows lying more than usual [195].

Pain can be assessed by using indirect measures such as gait and posture scoring on-farm. These are practical tools that require minimal equipment but require training and practice [197]. Locomotion scoring could also be assessed together with body condition scoring, leg hygiene, and hock condition. Winder et. al., have shown that dairy cow lameness caused an acute sole ulcer was considered painful by both veterinarians and producers, but veterinarians had a tendency to classify the pain level experienced by the cow slightly higher compared to producers [198]. This suggests that producers may need to be sensitized to pain occurring in their animals. There is an improved prognosis for lameness in cattle if it is diagnosed and treated at early stages [199]. Further, early detection of lameness minimizes animal suffering development of preventable associated health concerns, such as mastitis [200], and reduces the need for emergency culling, which can be associated with increased risk to the animal [22,174]. Foot rot and sole ulcers that are only diagnosed in early lactation are associated with a decreased survival rate [201]. Veterinarians and producers should develop a holistic diagnostic system to improve cattle health, welfare, and actions for affected animals [202].

Ethical decision-making and dilemmas: In severe cases of lameness, there are behavioral and locomotor changes, including gait abnormalities and increased pain-related lying that veterinarians and producers should be able to identify and use to determine a course of action. In some cases, euthanasia is the most appropriate treatment to reduce further suffering. Reaching this decision may be challenging because stakeholders may value the lame cow differently based on a different moral viewpoint of animal use and ownership. Heterogeneity in responses may arise in producers based on animal species, breed, and purpose [203]. Intensive breeding and farming systems often regard animals as commodities, which has been associated with reduced animal welfare. Ethical problems can be created when these severely affected cows have been identified by producers and transported to livestock sales barns [22]. In this scenario, there is an ethical dilemma if there is a conflict of interest between the veterinarian’s desire to treat and the producers’ approach to lameness in dairy cattle. The veterinarian-client relationship is vital to thoroughly expressing and understanding the potential welfare and ethical issues of underestimating, managing, and transporting animals with critical health issues, such as lameness. In most jurisdictions, veterinarians are considered the competent authority for determining whether an animal is fit for transport and regulations must be followed. Veterinarians must remain confident of their own judgments and knowledge, and when necessary, enlist the aid of producer and marketing association groups, as well as other stakeholder networks [204], which typically include regulatory authorities, to ensure appropriate treatment of animals on-farm.

## 6. Discussion

Animal welfare and sustainability issues arising from industrialized food animal production practices are ‘wicked problems’, which by definition are characterized as complex interconnected factors that can be defined and explained in a variety of ways, are unique, and are connected to other problems [205]. Wicked problems, by their very definition, have no single and definitive solution, and can only be effectively addressed when relevant stakeholders come together to engage in discussions about possible solutions. These may need to occur on a local, regional and national basis.

Ethical management of food animals is everyone’s responsibility—not just that of the veterinary practitioner. As nations struggle with growing populations, food security, and the devastating impact of climate change, sustainable practices for food animal production that respect animal integrity and welfare must be part of the discussion. For their part, to maintain relevancy, veterinarians must develop a willingness to question the status quo and challenge embedded and accepted animal practices. They must also be willing to reach out to industry stakeholders to find partners to engage with as well as new technologies to help support possible solutions.

## 7. Conclusions

Ethical concerns in industrial food animal production are increasing as societies in middle- and high-income countries are developing more awareness regarding how food is produced. Ethical aspects of food animal production are covered to some extent by legislation in some countries and regions, but more and more is being driven by NGO lobbying, public demand, and downstream market requirements.

Veterinarians could have an important role to play in managing many of the ethical dilemmas associated with industrial food animal production and can provide the knowledge that bridges many different opinions and stakeholders. However, current veterinary decision-making and participation may be hindered by limited training to ethical problem-solving, the constant economic conflict between advocating for improved animal care standards and maintaining client trust and making a living from practice, asynchronous legislative coverage of animal welfare even within the same country, and wide variations in societal concerns for specific food animal species, for example, laying hens vs meat rabbits or farmed fish. Despite this, there are several ethical frameworks that may aid veterinarians with decision-making and consensus building. The role of veterinarians in finding appropriate solutions can only increase if veterinarians are willing to tackle ethical challenges arising from the status quo in intensive food animal production systems. This is needed to address coming global changes for animals, people, and the environment.

## Figures and Tables

**Figure 1 animals-12-00678-f001:**
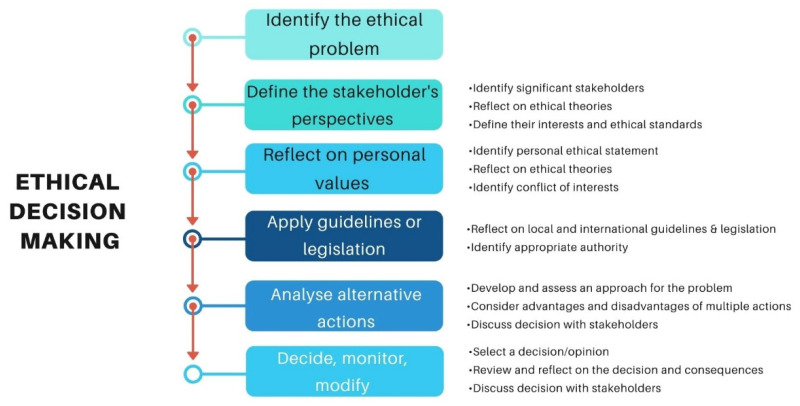
Common steps in veterinary ethical decision-making.

**Table 1 animals-12-00678-t001:** Variations in approaches to animal welfare legislation and recognition of animal sentience for the top three pork-producing countries, globally.

Pig Production Volume by Country and Associated Legislative Protection				
Rank	Country [36]	Tonnes of Meat, Pig [36]	Animal Welfare Legislation on Transport, Slaughter, and Rearing	Recognition of Animal Sentience and Veterinarian’s Role
1	China	41,133,300	The term “animal welfare” is not included in the present legislation [37] Transport: Animal Husbandry Law 2005 [38,39] Slaughter: Regulations on Administration of Hog Slaughter 1997 (revised 2007) [38] No legislation specifically on rearing of pigs [38,40]	Animal sentience not formally recognized [40] Veterinarians are not formally recognized as animal welfare advocates [41,42]
2	USA	12,845,097	Transport: Twenty-Eight hour Law [43,44] Slaughter: Humane Slaughter Act [43,45] No federal legislation specifically on rearing of pigs, but there are state by-laws [43]	Animal sentience is not formally recognized at the federal level, but there is legislation that recognizes suffering [43] Veterinarians are responsible for the animal health and welfare [46]
3	Germany	5,118,000	Transport: Council Directive EC 1/2005 [47] Slaughter: Council Directive EC No 1099/2009 [48] Rearing of pigs: Regulation on the Protection of Farm Animals at federal level, Council Directive 2008/120/EC at EU level [49,50,51]	Animal sentience is recognized in the Lisbon Treaty Veterinarians are responsible for animal health and welfare [52]

## Data Availability

Not applicable.

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
