# Peer review of "Applied Animal Ethics in Industrial Food Animal Production: Exploring the Role of the Veterinarian"

_animals, 2022, doi:10.3390/ani12060678_

Round 1
Reviewer 1 Report
In some ways this paper is an impressive piece of work – it covers a wide range of issues regarding industrial food animal production, farm animal welfare, different aspects of sustainability, animal ethics and veterinary ethics. However, unfortunately the paper completely lacks focus, lacks a clear disposition, contains inconsistencies, is in some ways extremely superficial, and contains a number of plain mistakes. Therefore, I think it should be rejected.
Let me elaborate a bit on these points:
Focus: The paper wanders into a number subjects, including for example the role of legislation and market driven initiatives which are interesting but are peripheral to the key theme of the paper, which is the role of the veterinarian. The lack of focus is clearly reflected in the abstract and conclusion which are both very vague, bordering on the trivial – cf. the last two sentences: “The role of veterinarians in finding appropriate solutions can only increase if veterinarians are willing to tackle ethical challenges arising from the status quo in food animal production. If this challenge can be met, veterinarians will continue to have a critical role in sustainable food animal production practices that will help to address coming global changes for animals, people, and the environment.”
Disposition: There is no clear line of thought connecting the four main sections – rather they more seem to be like separate, text-book like chapters.
Inconsistencies: Sometimes it seems that different bits of the texts are written by different authors who have not coordinated their efforts. For example in line 72-75 it is claimed that the modern concern for animal welfare started with the work of David Mellor and colleagues in the 1990ies and onwards: “The need to better balance human food security and animal production efficiency with animal welfare considerations is a more recent concern, as more mature models of animal welfare have been adopted, such as the five domains model”. However, line 189-195 it is said (correctly in my view) that the concern originated in the mid-1960ies: “The Brambell report gave both a definition of animal welfare and the first draft of the Five Freedoms, which were later modified by the Farm Animal Welfare Council. The Five Freedoms indicate that farm animals should have freedom from hunger and thirst, freedom from discomfort, freedom from pain, injury and disease, freedom to express normal behavior, and freedom from fear and distress [44]. More recently, the Five Freedoms have evolved into the concept of the Five Domains, described by Mellor , et al. …”
Another inconsistency concerns the discussion of veterinary and animal ethics. In the first sections of the paper several attempts are made to develop and defend alternatives to traditional pragmatic and consequentialist views on veterinary ethics, including virtue ethics, autonomy and justice. However, in the last main section where the theories are to be applied on a number of scenarios none of this comes up again and the analysis is very pragmatic and consequentialist. Again, it seems like different sections have been written completely independently of each other and have just been stitched together by means of a cut and paste maneuver.
Superficiality: The discussion of veterinary, animal and environmental ethics which is spread over several of the main sections is extremely superficial, bordering at name-dropping. Therefore a potential theoretical underpinning of a paper, allegedly on “applied animal ethics” is missing. Another example is the use of findings from consumer economics. For example, in line 656 it is claimed that “88% of consumers use animal welfare as a choice criterion”. Possibly this number can be found in the study mentioned. However, a brief look at the vast literature will reveal that this number must be an outlier.
Plain mistakes, here are three examples: 1) The position, utilitarian ethics, often mentioned, is a misnomer. What the authors have in mind is what the political theorist Robert Garner has called animal welfarism and deserves to be discussed as such. 2) In line 216-218 the authors mix up EU and the Council of Europe which are two independent entities – the mentioned conventions are by the Council and not the EU, and while the EU has taken over convention on farm animals, it has not taken over the convention on companion animals. 3) What is said in line 434-437 gives the (wrong) impression that currently there are GM animals in the food chain.
Author Response
Reviewer 1:
In some ways this paper is an impressive piece of work – it covers a wide range of issues regarding industrial food animal production, farm animal welfare, different aspects of sustainability, animal ethics and veterinary ethics. However, unfortunately the paper completely lacks focus, lacks a clear disposition, contains inconsistencies, is in some ways extremely superficial, and contains a number of plain mistakes. Therefore, I think it should be rejected.
Focus: The paper wanders into a number subjects, including for example the role of legislation and market driven initiatives which are interesting but are peripheral to the key theme of the paper, which is the role of the veterinarian.
We respectfully disagree with this comment. The role of legislation and market driven initiatives are, in fact, key constraints that significantly influence how veterinarians are able practice and exert influence. We have added text to better highlight these points at the end of the Introduction and throughout section 3. In Section 3 we have also added a table to emphasize the disconnect that often exists between IFAP, legislation and consideration of animal sentience, and support for a strong veterinary presence in the industry.
The lack of focus is clearly reflected in the abstract and conclusion which are both very vague, bordering on the trivial – cf. the last two sentences: “The role of veterinarians in finding appropriate solutions can only increase if veterinarians are willing to tackle ethical challenges arising from the status quo in food animal production. If this challenge can be met, veterinarians will continue to have a critical role in sustainable food animal production practices that will help to address coming global changes for animals, people, and the environment.”
Thank-you for your comment. We have significantly edited the Conclusion to emphasize the challenges and need for veterinarians to actively take on these ethical challenges.
Disposition: There is no clear line of thought connecting the four main sections – rather they more seem to be like separate, text-book like chapters.
Thank-you for your critique. We have added additional text throughout the paper to better conclude and link sections together to improve flow.
Inconsistencies: Sometimes it seems that different bits of the texts are written by different authors who have not coordinated their efforts. For example in line 72-75 it is claimed that the modern concern for animal welfare started with the work of David Mellor and colleagues in the 1990ies and onwards: “The need to better balance human food security and animal production efficiency with animal welfare considerations is a more recent concern, as more mature models of animal welfare have been adopted, such as the five domains model”. However, line 189-195 it is said (correctly in my view) that the concern originated in the mid-1960ies: “The Brambell report gave both a definition of animal welfare and the first draft of the Five Freedoms, which were later modified by the Farm Animal Welfare Council. The Five Freedoms indicate that farm animals should have freedom from hunger and thirst, freedom from discomfort, freedom from pain, injury and disease, freedom to express normal behavior, and freedom from fear and distress [44]. More recently, the Five Freedoms have evolved into the concept of the Five Domains, described by Mellor , et al. …”
Thank-you for your comment. We have introduced the Brambell Report in the Introduction, as suggested, to enhance consistency. This was an important first step for beginning to draw attention to food animal welfare, but the report and Five Freedoms were not broadly taken up within veterinary medicine until decades later.
Another inconsistency concerns the discussion of veterinary and animal ethics. In the first sections of the paper several attempts are made to develop and defend alternatives to traditional pragmatic and consequentialist views on veterinary ethics, including virtue ethics, autonomy and justice. However, in the last main section where the theories are to be applied on a number of scenarios none of this comes up again and the analysis is very pragmatic and consequentialist. Again, it seems like different sections have been written completely independently of each other and have just been stitched together by means of a cut and paste maneuver.
Thank-you for your comment. We have introduced a new figure to Section 5 that summarizes the ethical review process that is taken by veterinarians when exposed to an ethical dilemma, regardless of their ethical framework. While we introduce different ethical frameworks early in the paper, we emphasize that an applied bioethical approach is used in Section 5.We have also provided comments about a One Welfare ethical approach in Section 5 and what that might look like, as an alternative. We think a more practical approach will better engage veterinarians reading and thinking about these issues and how they might be able to act for change and this is also reflected in the title (applied animal ethics).
In addition, we have edited parts of Section 3, 4, and 5 to improve flow and grammar.
Superficiality: The discussion of veterinary, animal and environmental ethics which is spread over several of the main sections is extremely superficial, bordering at name-dropping. Therefore a potential theoretical underpinning of a paper, allegedly on “applied animal ethics” is missing. Another example is the use of findings from consumer economics. For example, in line 656 it is claimed that “88% of consumers use animal welfare as a choice criterion”. Possibly this number can be found in the study mentioned. However, a brief look at the vast literature will reveal that this number must be an outlier.
Thank-you for your comment. We agree that consumers may not act as virtuously as they indicate on a survey and have tempered these comments.
Plain mistakes, here are three examples: 1) The position, utilitarian ethics, often mentioned, is a misnomer. What the authors have in mind is what the political theorist Robert Garner has called animal welfarism and deserves to be discussed as such. 2) In line 216-218 the authors mix up EU and the Council of Europe which are two independent entities – the mentioned conventions are by the Council and not the EU, and while the EU has taken over convention on farm animals, it has not taken over the convention on companion animals. 3) What is said in line 434-437 gives the (wrong) impression that currently there are GM animals in the food chain.
Thank-you for these observations. We have left the terms ‘consequentialism’ and utilitarianism’ intact as these are common concepts that are taught to most veterinarians whereas ‘animal welfarism’ has received limited uptake.
The EU Council error has been corrected – thank-you.
We have clarified the comment regarding GM animals to indicate that this occurs in research settings.
Reviewer 2 Report
I thank the authors for submitting a well crafted and thoughtful manuscript. A very stimulating document which, I hope, will prompt the discussion and development of the role of the veterinarian from a perspective of one welfare. There are little if any corrections to be made, beyond some minor ones noted below.
Line 94: [10] .
Line 122: reference 26 doesn’t seem proper
Line 348: Reference 78, please add an updated reference.
Row 373-374: This conclusion should be more substantiated.
Line 391-392: “.. as mentioned previously” sorry, but I didn’t find where you mentioned
Line 464-469: “.. that may raise ethical concerns viewed from some stakeholders’ perspectives (e.g., precision livestock farming).” Please clarify
Line 602-617: I believe you should insert this paragraph in the introductory section.
Line 628: “Some degree of intensification in animal agriculture”, I would change the word intensification
Line 741: [references: 153,157] here you can find more recent data: https://www.ema.europa.eu/en/veterinary-regulatory/overview/antimicrobial-resistance/european-surveillance-veterinary-antimicrobial-consumption-esvac#trends-by-country-section
You can check, also, the example of Italy, Italy adopted a computerised traceability system for the veterinary supply chain, which has allowed a decrease in the use of antibiotics.
Line 792-793: please add a reference
Discussion: I would add that there is a need for financial assistance to farmers.
Author Response
Reviewer 2:
I thank the authors for submitting a well crafted and thoughtful manuscript. A very stimulating document which, I hope, will prompt the discussion and development of the role of the veterinarian from a perspective of one welfare. There are little if any corrections to be made, beyond some minor ones noted below.
Line 94: [10] .
Corrected, thank-you.
Line 122: reference 26 doesn’t seem proper
This reference was used to demonstrate the disconnect between the theoretical knowledge learned during vet education (in this example, about animal sentience and the need for routine pain management for painful practices vs how practitioners act in the field when clients push back on costs or other factors). Unfortunately, practitioners do not routinely apply animal welfare principles at all times. There is very little information available that examines what vets know vs how they practice when it comes to animal welfare.
Line 348: Reference 78, please add an updated reference.
This is a 2015 reference and, respectfully, we believe that the content is still relevant.
Row 373-374: This conclusion should be more substantiated.
Thank-you for this comment. The content and references in the paragraph immediately preceding this one provide the evidence for the summary comments. This emphasizes again that welfare legislation and practices as they pertain to food animals are asynchronous. We have attempted to tie this idea together better by adding more summary sentences throughout section 3.
Line 391-392: “.. as mentioned previously” sorry, but I didn’t find where you mentioned
Thank-you for pointing this out. The sentence referred to content that had been deleted in the original draft of the MS. This sentence has also been deleted.
Line 464-469: “.. that may raise ethical concerns viewed from some stakeholders’ perspectives (e.g., precision livestock farming).” Please clarify
We have added additional information and a reference to clarify, as requested.
Line 602-617: I believe you should insert this paragraph in the introductory section.
Thank-you for this suggestion. Respectfully, after discussion by the authors, we did decide to leave it here because it is highly pertinent to this scenario, but we have added content to Section 3 that highlights this conflict.
Line 628: “Some degree of intensification in animal agriculture”, I would change the word intensification
As requested, we have changed this to ‘some industrialized aspects of food animal management’.
Line 741: [references: 153,157] here you can find more recent data: https://www.ema.europa.eu/en/veterinary-regulatory/overview/antimicrobial-resistance/european-surveillance-veterinary-antimicrobial-consumption-esvac#trends-by-country-section
You can check, also, the example of Italy, Italy adopted a computerised traceability system for the veterinary supply chain, which has allowed a decrease in the use of antibiotics.
Thank-you for the suggestion. We have modified the text in this section and added additional references.
Line 792-793: please add a reference
A reference has been added, as suggested.
Discussion: I would add that there is a need for financial assistance to farmers.
Respectfully, adding content to discuss how better to support producers to make change was thought to be outside of the scope of this paper and how this is done will vary widely between global regions (i.e., not all countries are able to provide farmer subsidies). We have tried to focus on ‘actionable items’ for veterinarians.
Reviewer 3 Report
This a very interesting paper, clear in concepts and well written, in which the authors review the traditional veterinary ethical approach to industrial food animal production and pertinent animal welfare legislation for industrial food animal production. They also give some examples on how veterinarians can provide leadership in improving food animal welfare by supporting changes in animal housing and management practices. It is important for veterinarians to adopt new views in relation to farm animal production and include animal welfare and environmental sustainability aspects in their daily work, also transmitting these issues to the producers. The consequences of disregarding animal welfare are well explained in terms of animal welfare and also “quantity” of products (decreases in productivity), however, I would have liked to see a bit more about the relationship between animal welfare, proper handling and reduced stress on the quality of food products (somatic cell counts in milk or inadequate pH and bruises in meat of various species) . This can also be a driver towards better animal welfare and could be added, to make veterinarians aware of these consequences as well. In the case of the use of antibiotics, this is analyzed in depth.
Some minor comments follow:
219 “…of the Amsterdam treaty and the Five Conventions, “….there is no reference or year for this
365-367 “ In addition, there is a joint effort from the veterinary schools in Chile, Uruguay, and Mexico to support the OIE Collaborating Centre for Animal Welfare Research and Livestock Production Systems. “ I would like to clarify that there is a joint effort from the veterinary schools in all Latin America, and a commitment, to include animal welfare in the veterinary curriculum. Chile, Uruguay, and Mexico form the Collaborating Centre for AW and promote AW in many aspects, besides the teaching (as the authors themselves state further below in the same paragraph)
482-486 “There is an opportunity for veterinarians to bring the animal health and welfare perspective into the debate and clarify controversial animal welfare issues (e.g., zoonotic diseases, food security, antimicrobial reduction, etc.) and there is an opportunity for veterinarians to provide the animal health and welfare perspective and clarify controversial animal welfare issues”. Please revise this sentence, as it seems concepts are repeated…
523 “ The includes an optimization…”, This includes
621….”Methane is extracted during enteric fermentation…”, is it not produced instead of extracted?
797….”A One Wealth approach….”, One Health?
825..”… for more disease preventative solutions in the poultry…., preventive?
- “….situations with producers in which it not easy to provide…”, please correct “it is not easy”
- “There is an improved prognosis for lameness in cattle if it is diagnosed and treated at early stages [185]”. In regard to this statement, it would be important to add that an early diagnosis could also help in taking an earlier decision to cull, and then the cow benefits from not being in pain so long and the producer also benefits from selling a cow that is not yet in a severely low body condition. Perhaps comment more on these “late decisions” for culling, that affect both the animal and the producer and which finally may even produce carcass condemnation, i.e. total loss of product.
874-877. “ In some cases, euthanasia is the most appropriate solution to reduce further pain and discomfort if there is no treatment available. Reaching this decision may be challenging because stakeholders may value the lame cow differently based on a different moral viewpoint of animal use and ownership.” It would be useful to add some existing guidelines on “when” and “how” to do euthanasia (broadly, how to make the decision).
Author Response
Reviewer 3:
This a very interesting paper, clear in concepts and well written, in which the authors review the traditional veterinary ethical approach to industrial food animal production and pertinent animal welfare legislation for industrial food animal production. They also give some examples on how veterinarians can provide leadership in improving food animal welfare by supporting changes in animal housing and management practices. It is important for veterinarians to adopt new views in relation to farm animal production and include animal welfare and environmental sustainability aspects in their daily work, also transmitting these issues to the producers. The consequences of disregarding animal welfare are well explained in terms of animal welfare and also “quantity” of products (decreases in productivity), however, I would have liked to see a bit more about the relationship between animal welfare, proper handling and reduced stress on the quality of food products (somatic cell counts in milk or inadequate pH and bruises in meat of various species) . This can also be a driver towards better animal welfare and could be added, to make veterinarians aware of these consequences as well. In the case of the use of antibiotics, this is analyzed in depth.
Thank-you for this suggestion. We have added information in section 3.5 to address this in the context of the importance of voluntary food animal welfare assurance scheme participation for driving change.
Some minor comments follow:
219 “…of the Amsterdam treaty and the Five Conventions, “….there is no reference or year for this
A reference and year have been added, as requested.
365-367 “ In addition, there is a joint effort from the veterinary schools in Chile, Uruguay, and Mexico to support the OIE Collaborating Centre for Animal Welfare Research and Livestock Production Systems. “ I would like to clarify that there is a joint effort from the veterinary schools in all Latin America, and a commitment, to include animal welfare in the veterinary curriculum. Chile, Uruguay, and Mexico form the Collaborating Centre for AW and promote AW in many aspects, besides the teaching (as the authors themselves state further below in the same paragraph)
Thank-you for this suggestion. We have modified the sentence to be more inclusive.
482-486 “There is an opportunity for veterinarians to bring the animal health and welfare perspective into the debate and clarify controversial animal welfare issues (e.g., zoonotic diseases, food security, antimicrobial reduction, etc.) and there is an opportunity for veterinarians to provide the animal health and welfare perspective and clarify controversial animal welfare issues”. Please revise this sentence, as it seems concepts are repeated…
Revised to remove the redundancy, as suggested.
523 “ The includes an optimization…”, This includes
Corrected, as noted, thank-you.
621….”Methane is extracted during enteric fermentation…”, is it not produced instead of extracted?
Corrected, as noted – thank-you.
797….”A One Wealth approach….”, One Health?
Corrected typo – thank-you.
825..”… for more disease preventative solutions in the poultry…., preventive?
Corrected, as suggested – thank-you.
- “….situations with producers in which it not easy to provide…”, please correct “it is not easy”
Corrected grammar – thank-you.
- “There is an improved prognosis for lameness in cattle if it is diagnosed and treated at early stages [185]”. In regard to this statement, it would be important to add that an early diagnosis could also help in taking an earlier decision to cull, and then the cow benefits from not being in pain so long and the producer also benefits from selling a cow that is not yet in a severely low body condition. Perhaps comment more on these “late decisions” for culling, that affect both the animal and the producer and which finally may even produce carcass condemnation, i.e. total loss of product.
Additional information added, as suggested.
874-877. “ In some cases, euthanasia is the most appropriate solution to reduce further pain and discomfort if there is no treatment available. Reaching this decision may be challenging because stakeholders may value the lame cow differently based on a different moral viewpoint of animal use and ownership.” It would be useful to add some existing guidelines on “when” and “how” to do euthanasia (broadly, how to make the decision).
Additional general guidance has been added to increase confidence in decision-making, as suggested.